# Effect of Ultrasonic Pretreatment on the Extraction Process of Essential Oils from Grapefruit (*Citrus paradisi*) By-Products

**DOI:** 10.3390/biotech14030059

**Published:** 2025-08-07

**Authors:** Francisco Cadena-Cadena, Joe Luis Arias-Moscoso, Leandris Argentel-Martínez, Jony R. Torres Velazquez, Dulce Alondra Cuevas-Acuña, Nydia Estrellita Buitimea Cantua, Bartolo Concha-Frías

**Affiliations:** 1Departamento de Ingeniería, Instituto Tecnológico del Valle del Yaqui, Ciudad Obregón CP 85150, Sonora, Mexico; fcadena.cadena@itvy.edu.mx (F.C.-C.); oleinismora@gmail.com (L.A.-M.); jtorres.velazquez@itvy.edu.mx (J.R.T.V.); 2Departamento de Ciencias de la Salud, Universidad de Sonora, Ciudad Obregón CP 85150, Sonora, Mexico; 3Departamento de Investigación y Posgrado en Alimentos, Universidad de Sonora, Hermosillo CP 83000, Sonora, Mexico; nydia.buitimea@unison.mx; 4Laboratory of Nutrigenomics and Toxicology of Native Fish, Division of Basic Sciences and Engineering, Universidad Popular de la Chontalpa, Cárdenas CP 86529, Tabasco, Mexico

**Keywords:** functional properties, by-products, grapefruit, ultrasound pulses

## Abstract

This study evaluated the effect of ultrasonic pulse-assisted extraction on the yield and antioxidant activity of essential oils from grapefruit (*Citrus paradisi*) by-products using hydrodistillation and Soxhlet solvent extraction (hexane, acetone, ethanol). Ultrasound was applied at 40% amplitude for 20 min before extraction. Results showed that ultrasound significantly increased extraction yield with hexane (from 2.6 ± 0.58% to 7.6 ± 1.5%) and acetone (from 8.6 ± 0.96% to 12 ± 1.4%), while ultrasound-assisted hydrodistillation nearly doubled the yield (from 0.7 ± 0.03% to 1.5 ± 0.49%). In contrast, ultrasound decreased yield with ethanol by 3%. Antioxidant activity measured by TEAC assay was highest in acetone extracts without ultrasound (13,366.5 ± 7.66 mmol TE/g) and ethanol extracts (12,606.8 ± 0.51 mmol TE/g). However, ultrasound combined with ethanol increased DPPH scavenging activity from 1073.5 ± 1.07 µg/mL to 4933.3 ± 0.71 µg/mL and maintained high flavonoid content (9.41 ± 0.15 mg/mL) and phenolics (5.33 ± 0.09 mg/mL). Ultrasound-assisted hydrodistillation also enhanced antioxidant capacity, with DPPH values rising from 51.82 ± 5.56 µg/mL to 2413.03 ± 3.17 µg/mL. These findings demonstrate that ultrasound effectively enhances essential oil extraction and antioxidant activity depending on the solvent used, underscoring the potential of this clean technology for valorizing citrus by-products.

## 1. Introduction

Citrus fruits, such as grapefruit (*Citrus paradisi*), are a natural and accessible source of antioxidant and antimicrobial compounds, which are highly valuable to the pharmaceutical and food industries [1]. Notably, grapefruit residues, such as peels and seeds, represent a waste material with great potential for the extraction of bioactive compounds, including essential oils and pectin, a dietary fiber powder [2,3,4].

Various techniques are available for extracting bioactive and functional compounds from grapefruit by-products, such as microwave and hydrodistillation for essential oil extraction [5], ultra force grinding, and microwave-assisted extraction [6,7] for dietary fiber extraction, and ultrasound, ohmic heating, and microwave-assisted high pressure for pectin extraction [8,9,10].

Ultrasound-assisted extraction has gained considerable attention due to its high efficiency. It employs ultrasonic waves to break down cell walls and facilitate the release of bioactive compounds, thus improving extraction yield [11,12]. These techniques enable the extraction of essential oils with a quality profile that preserves the desired aromatic and functional properties for use in the food and cosmetic industries [13,14].

Ultrasonic pulses are acoustic waves with frequencies above the human hearing threshold, i.e., above 16 kHz. These waves propagate through a material at a specific speed, which depends on the material’s physical and chemical properties. Ultrasonic pulses are widely used in industrial and scientific applications such as compound extraction, surface cleaning, and material characterization [15].

Pulsed ultrasound waves generate mechanical energy that produces high-intensity shear forces, facilitating the rupture of structures and the destruction of cell walls. This phenomenon enables the more efficient extraction of target compounds, such as antioxidants, essential oils, and other metabolites [16,17].

Previous studies on essential oils extracted from grapefruit peel using hydrodistillation and microwave methods have shown that they are mainly composed of limonene and possess bacteriostatic, but not bactericidal, properties, against a broad spectrum of bacteria [5,18]. Other studies on grapefruit peel extracts, also extracted by hydrodistillation, have shown antioxidant activity in in vitro tests [19].

However, although several studies have been conducted on the extraction of essential oils from citrus peels, most have focused on oranges, and those on grapefruit essential oils utilize ultrasonic baths [20]. Extraction studies comparing the ultrasonic bath method with an ultrasonic probe demonstrate greater extraction effectiveness with the latter [21].

Therefore, this research aims to evaluate the effect of ultrasonic pulses, using a probe-type sonicator as pretreatment, on the yield and antioxidant properties of essential oils extracted from grapefruit by-products.

## 2. Materials and Methods

This study was conducted in three main stages: (I) conditioning of the raw material (grapefruit by-products) to determine the optimal extraction conditions; (II) essential oil extraction using hydrodistillation and Soxhlet solvent extraction, both assisted by ultrasound; and (III) chemical–proximate composition analysis of the peel and evaluation of the antioxidant activity of the extracted oils.

### 2.1. Raw Material Conditioning

Grapefruit by-products were first examined, and any material with significant damage to the peel, such as punctures, signs of rot, or over-ripeness, was discarded. The remaining pulp, seeds, and albedo were also removed in order to retain only the portion of interest [22].

The peel obtained as a by-product of grapefruit was carefully washed to remove impurities and then cut into sections of approximately 2 to 3 cm. These pieces were placed in a drying oven at 60 °C for 24 h, a temperature chosen to prevent degradation or denaturation of the essential oils [23].

### 2.2. Extraction Methods

Essential oils were extracted using hydrodistillation and the Soxhlet solvent extraction method based on the procedures described by Colina-Marquez et al. [24] and Chemat et al. [25], with some modifications. In both methods, ultrasonic pulse assistance was applied.

### 2.3. Hydrodistillation Extraction

Distillation was carried out using water to extract the essential oils from the grapefruit peel. Approximately 180 g of raw material was used for a 3 h extraction, ensuring the temperature did not exceed 90 °C, following the method of [24].

### 2.4. Solvent Extraction

In this process, hexane, acetone, and ethanol were used as solvents. They were selected for their polarity range and affinity for extracting lipophilic and polar fractions from essential oils. Hexane, a non-polar solvent, solubilizes highly hydrophobic compounds. Acetone, an intermediate polar solvent, allows the recovery of moderately polar metabolites. Ethanol, a polar and renewable solvent, aligns with the principles of green chemistry because it is less toxic and biodegradable [25,26]. For each treatment, 8 g of dry, ground peel was placed in a filter paper cartridge and inserted into a Soxhlet apparatus containing 250 mL of the corresponding solvent. The extraction temperature was adjusted according to the properties of each solvent, following the methodology described by [24].

### 2.5. Ultrasound-Assisted Extraction

Prior to hydro or solvent extraction, grapefruit by-product samples were subjected to ultrasonic pulses. A high-intensity probe connected to a Sonics Ultra Cell VCX-750 ultrasonic processor was used. Samples were mixed at a 1:6 ratio with each solvent, according to the method proposed by [27]. The ultrasonication conditions were as follows: 20-min treatment, 40% amplitude, with 20 s on/10 s off cycles, using a power of 750 W (Figure 1). The selected conditions were based on preliminary studies analyzing various combinations of time, amplitude, and cycles. These studies aimed to maximize cell rupture efficiency, promote the release of bioactive compounds, and minimize thermal degradation. These parameters are also consistent with values reported in the literature for comparable plant matrices by Jagannath et al. [27], including oranges (Hybrid King) and grapefruit [27,28].

### 2.6. Chemical–Proximate Composition and Antioxidant Activity Evaluation

#### Proximate Composition

▪The proximate composition of the grapefruit peel was determined using the official methods described by [29], as follows:▪Moisture (Method 925.09): calculated as weight loss after drying 2 g of sample in an oven at 110 °C for 2 h.▪Ash (Method 923.03): determined by weighing the residue after incinerating the sample in a muffle furnace at 550 °C for 2 h.▪Protein (Method 2001.11): determined by digesting 0.1 g of defatted sample in H_2_SO_4_, followed by distillation and nitrogen recovery in boric acid, and titration with 0.1 N NaOH.▪Fat (Method 920.39): determined by continuous solvent extraction using petroleum ether in a Soxhlet system.

### 2.7. Antioxidant Activity Evaluation

Antioxidant capacity was evaluated using DPPH and TEAC assays. Results are presented as mean ± standard deviation (SD) from three independent replicates. To determine significant differences among extraction treatments, a one-way analysis of variance (ANOVA) was conducted, followed by Tukey’s honestly significant difference (HSD) post hoc test for pairwise comparisons at a significance level of *p* < 0.05.

DPPH assay: antioxidant activity was determined using the DPPH (2,2-diphenyl-1-picrylhydrazyl) method, with modifications to the protocol by [30]. A volume of 0.1 mL of extract was mixed with 2.9 mL of pure methanol. After adding 0.5 mL of DPPH solution, the mixture was incubated in the dark at room temperature for 45 min. Absorbance was measured at 515 nm using a Thermo Scientific Multiskan Sky microplate reader. A blank (3 mL methanol + 0.5 mL DPPH) and control (3 mL methanol) were used. Antioxidant activity was expressed as the percentage of DPPH inhibition using the following formula:FRSA = 100 × (initial absorbance − final absorbance)/initial absorbance.

TEAC assay: the Trolox equivalent antioxidant capacity (TEAC) assay was performed according to [31]. Antioxidant activity was expressed as millimoles of Trolox equivalents per gram of extract (mmol TE/g extract), and the percentage of inhibition (%) was also calculated.

Total phenolics: a 1 mg/mL gallic acid stock solution was used to construct a calibration curve via serial dilutions. Twenty microliters of extract or standard (in triplicate) were mixed with 1.5 mL of distilled water and 100 μL of Folin–Ciocalteu reagent. After 5 min, 300 μL of 20% sodium carbonate solution was added, and the mixture was incubated for 2 h at room temperature (or 30 min at 40 °C). Absorbance was read at 415 nm. Results were expressed as mg gallic acid equivalents per mL.

Total flavonoids: the colorimetric method described by Zhishen et al. [32] and Dewanto et al. [30] was used, with a 0.1 mg/mL rutin stock solution and appropriate dilutions for the calibration curve. One hundred microliters of extract or standard (in triplicate) were mixed with 1 mL of methanol and 50 μL of 1% 2-aminoethyl diphenylborinate. After 30 min at room temperature, absorbance was measured at 765 nm.

### 2.8. Experimental Design and Statistical Analysis

A factorial experimental design was employed to evaluate the impact of ultrasound assistance, solvent type, and extraction method on essential oil extraction yield and antioxidant activity from grapefruit peel. The following factors were tested: extraction method (hydrodistillation and Soxhlet solvent extraction); solvent type for Soxhlet extraction (hexane, acetone, and ethanol); and ultrasound assistance (with vs. without ultrasound pretreatment).

Each treatment combination was performed in triplicate (n = 3), resulting in robust data for statistical analysis.

The outcomes measured included: extraction yield (%); antioxidant activity using TEAC, DPPH assays, total flavonoids, and total phenolics.

Data normality and homogeneity of variances were verified using the Shapiro–Wilk and Levene tests, respectively. A three-way ANOVA was conducted to determine the significance of main effects and interactions, followed by Tukey’s HSD post hoc test (*p* < 0.05). Effect sizes (η^2^) were calculated to quantify the contribution of each factor to the observed variability in the outcomes. Statistical analysis was performed using JMP 9 software.

## 3. Results and Discussion

### 3.1. Extraction Yield

The extraction yields varied significantly depending on the solvent and the application of ultrasonic assistance (Table 1). Ultrasonic-assisted treatments using hexane (U1) and acetone (U3) consistently demonstrated higher yields compared to their non-assisted counterparts (E1 and E3). This enhancement can be attributed to the cavitation effect, where microscopic bubbles generated during sonication collapse, producing localized pressure and temperature gradients that disrupt cell walls and improve solvent penetration [33,34].

However, ethanol extraction (E2) exhibited an unexpected trend: the highest yield (20%) was achieved without ultrasonic assistance, whereas sonication (U2) reduced the yield by 3%. This reduction may be due to ethanol’s high polarity and susceptibility to sonochemical degradation [35]. The energy released during cavitation could destabilize ethanol molecules or degrade thermally sensitive intermediates [36,37]. Additionally, ethanol’s strong affinity for polar compounds (e.g., waxes, gums) may lead to co-extraction of non-target compounds, increasing yield but potentially reducing extract purity [38,39].

Non-polar solvents (hexane, acetone) produced extracts with higher purity, as they selectively solubilize non-polar essential oils [39]. Ultrasonication further improved yields by enhancing volatile compound extraction via cavitation-driven matrix disruption. Ethanol, being highly polar, extracted additional polar constituents (e.g., polyphenols, pigments), which may explain its higher yield but lower purity [40].

The ultrasound-assisted hydrodistillation (A2) yielded 1.5%—double that of conventional hydrodistillation (A1, 0.7%). This dramatic improvement underscores ultrasonication’s role in rupturing oil glands and enhancing mass transfer before distillation [17]. The cavitation-induced mechanical effects likely created micro-fractures in the plant matrix, facilitating faster and more complete essential oil release [18].

While ultrasonication generally enhances extraction yield by facilitating cell wall rupture and improving mass transfer, it may also co-extract non-volatile and polar compounds, potentially affecting extract purity [33]. Ethanol, due to its high polarity, can efficiently solubilize a broad range of polar compounds, including waxes, gums, mucilage, and cellulose, leading to higher extraction yields [29]. However, when combined with ultrasonication, cavitation may further promote the solubilization of these undesirable residues, which can reduce the purity of the essential oil fraction [33,34].

The localized high temperatures and pressures generated during cavitation can also contribute to the degradation of ethanol-soluble compounds or the ethanol itself, potentially reducing extraction efficiency [31]. Additionally, the collapse of microbubbles during cavitation releases localized energy, which may destabilize sensitive compounds and cause their volatilization or degradation, further influencing yield and extract composition [32].

In contrast, non-polar solvents such as hexane typically produce extracts with higher purity but lower overall yields, as they selectively extract volatile and non-polar compounds while excluding many polar impurities [34,35]. When ultrasonic assistance is applied with hexane, a notable increase in yield can be observed due to cavitation facilitating the release of volatile compounds from the plant matrix, while maintaining relatively high purity [34].

Overall, ultrasonic assistance can significantly enhance extraction yields, particularly with non-polar and moderately polar solvents, by generating expansion cycles and collapsing bubbles that create negative pressure, facilitating the rupture of the peel and exposing oil glands containing essential oils [27]. However, with highly polar solvents like ethanol, ultrasonication may have a dual effect: while it can enhance extraction, it may also increase the co-extraction of polar impurities and promote compound degradation, requiring careful optimization of extraction conditions to balance yield and purity.

### 3.2. Proximate Composition Determination

Table 2 presents the proximate composition of grapefruit peel, compared with values reported for other citrus species such as mandarin and orange. The results obtained in this study for grapefruit peel show protein, lipid, moisture, and ash contents consistent with those reported by [41] for the same species. However, differences were observed in the moisture and ash percentages, which may be attributed to variations in cultivation conditions, fruit maturity, or analytical methods [42,43].

When compared with other citrus fruits, grapefruit peel showed a lower protein content than mandarin (7.55% ± 0.24) and a similar value to that of orange (6.00%). Regarding lipid content, grapefruit peel had a higher value than mandarin (1.45% ± 0.16) and was comparable to that of orange (3.40%). The moisture content obtained in this study (7.17% ± 0.65) was similar to the value reported by [41] for grapefruit (7.81% ± 0.10), but significantly lower than the value recorded by [44] (62.55%). This discrepancy may be due to differences in sample processing or the degree of dehydration of the analyzed peels [45].

As for ash content, grapefruit peel in this study (5.13% ± 0.46) showed higher values than those reported by [46] (2.99% ± 0.20) and [24] for orange (3.92%), but lower than the values reported by [47] for orange (6.90%). These variations may be related to differences in soil mineral composition or extraction and the analytical methods used [48].

**Table 2 biotech-14-00059-t002:** Proximate composition of grapefruit peel.

Sample	%Protein	%Lipids	%Moisture	%Ash	Reference
Mandarin	7.55 ± 0.24	1.45 ± 0.16	4.33 ± 0.07	3.96 ± 0.21	[49]
Grapefruit	4.22 ± 0.25	2.01 ± 0.10	7.81 ± 0.10	2.99 ± 0.20	[49]
	-	-	62.55	5.09	[50]
Orange	6.00	3.40	-	6.90	[38]
	1.96	2.2	-	3.92	[51]
Grapefruit	4.5 ± 0.34	3.6 ± 1.2	7.17 ± 0.65	5.13 ± 0.46	Present research

Values represent the mean of 3 replicates ± standard deviation.

### 3.3. Antioxidant Activity

The results presented in Table 3 show significant variations in antioxidant activity among the different treatments, as evaluated by TEAC, DPPH, total flavonoids, and total phenolics assays. It is important to note that antioxidant activity is directly related to the concentration of the extract. All values are expressed as the mean of three replicates ± standard deviation and were calculated in Trolox equivalents (TEAC) for the essential oil samples analyzed.

### 3.4. Hydrodistillation

Sample A2 (hydrodistillation with ultrasound assistance) showed a significant increase in antioxidant capacity compared to A1 (without ultrasound). In the DPPH assay, A2 reached 2413.03 ± 3.17 µg/mL, whereas A1 recorded only 51.82 ± 5.56 µg/mL. This suggests that ultrasonic pulses enhance the efficiency of antioxidant compound extraction, likely due to cavitation, which facilitates cell disruption and the release of bioactive molecules [15,52].

Similarly, in total phenolic content, A2 reached 1.19 ± 0.07 mg/mL compared to A1’s 0.96 ± 0.04 mg/mL, further supporting the hypothesis that ultrasonic pulses optimize the extraction of these compounds [48].

### 3.5. Solvent Extraction

Among the solvents used, ethanol (E2) and acetone (E3) produced the highest antioxidant capacities across all tests. In the TEAC assay, E3 yielded 13,366.5 ± 7.66 mmol TE/g, followed by E2 with 12,606.8 ± 0.51 mmol TE/g. These results indicate that these polar solvents are more effective in extracting antioxidant compounds than hexane (E1), which showed significantly lower values (303.2 ± 3.26 mmol TE/g).

In the DPPH assay, E3 again performed best (4363.9 ± 4.14 µg/mL), followed by E2 (1073.5 ± 1.07 µg/mL), reaffirming that solvent polarity plays a crucial role in the extraction of antioxidant compounds.

### 3.6. Ultrasound-Assisted Solvent Extraction

The application of ultrasonic pulses during solvent extraction produced mixed results. In the case of ethanol with ultrasound (U2), a significant increase in DPPH activity (4933.33 ± 0.71 µg/mL) was observed compared to ethanol without ultrasound (E2: 1073.5 ± 1.07 µg/mL). This confirms the enhancing effect of ultrasound on the extraction of antioxidant compounds in ethanol.

However, in the case of acetone with ultrasound (U3), a decrease in antioxidant activity was observed compared to acetone alone (E3). In the TEAC assay, U3 yielded only 641.0 ± 1.40 mmol TE/g, in contrast with E3’s much higher 13,366.5 ± 7.66 mmol TE/g. This may be due to degradation or instability of antioxidant compounds caused by the combined effects of ultrasound and acetone [53].

The application of ultrasound induces acoustic cavitation, which generates transient zones of high temperature and pressure. These zones can promote the release and degradation of bioactive compounds [54]. Due to its intermediate polarity, acetone can facilitate the extraction of certain phenolic compounds and flavonoids. However, its ability to protect these compounds from free radicals generated by cavitation, such as hydroxyl radicals (•OH), is limited [55]. This could explain the significant decrease in antioxidant activity in treatment U3.

In contrast, ethanol is highly efficient at solubilizing antioxidant compounds and can act as a moderate radical scavenger, reducing the degradation of metabolites extracted during sonication [15]. The differences in the interaction between solvents and cavitation could explain the disparity between U2 and U3. Thus, the extraction efficiency and chemical stability of bioactive compounds under extreme processing conditions must be considered. Evidence suggests that while ultrasound is an effective tool for improving extraction yield, its application must depend on the type of solvent used and the thermal sensitivity of the target metabolites. Previous studies have reported that antioxidants, such as certain phenolic acids, glycosylated flavonoids, and volatile compounds, can degrade under intense cavitation conditions, especially when using less protective solvents [56]. Therefore, the selection of the solvent-ultrasound system must be based on extraction criteria and molecular stability considerations.

### 3.7. Total Flavonoids and Phenolics

The trends observed in total flavonoids and phenolics were consistent with those in the TEAC and DPPH assays. For example, ethanol with ultrasound (U2) showed high levels of flavonoids (9.41 ± 0.15 mg/mL) and phenolics (5.33 ± 0.09 mg/mL), supporting its strong antioxidant capacity.

In contrast, hexane with ultrasound (U1) showed low values for both flavonoids (1.64 ± 0.17 mg/mL) and phenolics (0.80 ± 0.05 mg/mL), confirming its limited ability to extract bioactive compounds.

In general, the application of ultrasonic pulses improved antioxidant capacity in most treatments, particularly with hydrodistillation and polar solvents like ethanol. However, the effect of ultrasound may vary depending on the solvent used, as in some cases it may reduce antioxidant activity due to possible degradation processes. These findings highlight the importance of optimizing extraction conditions, considering both the type of solvent and the use of auxiliary technologies such as ultrasound pulses.

## 4. Conclusions

This study has demonstrated that ultrasonic pulse-assisted extraction significantly enhances the yield and antioxidant activity of essential oils obtained from grapefruit (*Citrus paradisi*) by-products. The highest extraction yields were achieved using acetone and hexane under ultrasonic assistance, while ethanol without ultrasound provided the highest yield overall.

Ultrasound-assisted hydrodistillation nearly doubled the extraction yield compared to the conventional method, highlighting the potential of this technology to improve the efficiency of essential oil recovery from agro-industrial residues.

Regarding antioxidant activity, polar solvents such as ethanol and acetone were more effective than non-polar solvents, and ultrasonic assistance generally improved the extraction of antioxidant compounds. However, the interaction between ultrasound and solvent type proved critical: while ultrasound enhanced ethanol-based extractions, it appeared to degrade antioxidant compounds when combined with acetone.

These findings underscore the importance of tailoring extraction protocols according to solvent polarity and technological assistance. The use of ultrasound presents a promising, environmentally friendly alternative for valorizing citrus by-products and obtaining high-value bioactive compounds.

This study has demonstrated that ultrasound-assisted extraction enhances the yield and antioxidant activity of essential oils from grapefruit by-products, offering a sustainable and efficient valorization strategy. However, limitations include the lack of detailed chemical profiling of the extracts and scalability assessments under industrial conditions. Future studies should focus on compositional analyses, evaluating continuous-flow ultrasound systems, and testing the bioactivity of the extracted oils in food and cosmetic applications to fully realize their functional and commercial potential.

## Figures and Tables

**Figure 1 biotech-14-00059-f001:**
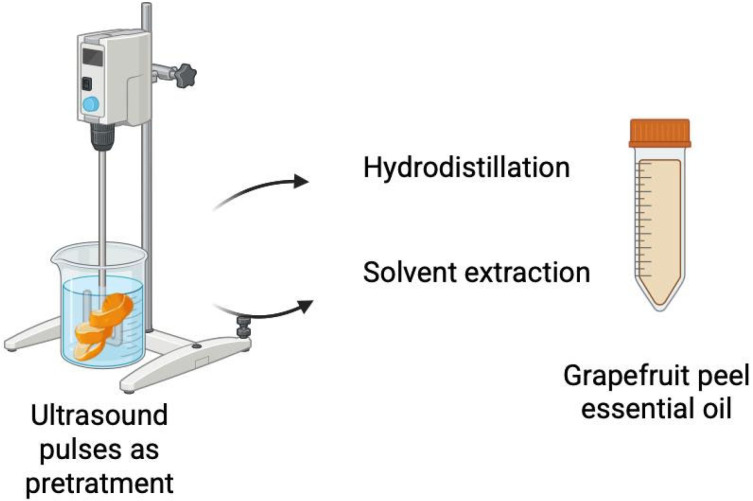
Schematic representation of the ultrasound-assisted extraction experimental setup.

**Table 1 biotech-14-00059-t001:** Essential oil extraction yield (expressed as %).

Treatment	*Citrus paradisi*
E1	2.6 ± 0.58
U1	7.6 ± 1.5
E2	20 ± 2.7
U2	17 ± 1.8
E3	8.6 ± 0.96
U3	12 ± 1.4
A1	0.7 ± 0.03
A2	1.5 ± 0.49

Treatments: E1—hexane; U1—hexane + ultrasound; E2—ethanol; U2—ethanol + ultrasound; E3—acetone, U3—acetone + ultrasound; A1—hydrodistillation; A2—hydrodistillation + ultrasound. Values are the mean of three replicates ± standard deviation.

**Table 3 biotech-14-00059-t003:** Antioxidant activity of essential oil extracts.

Treatments	TEAC mmol TE/g	DPPH µg/mL	Flavonoides mg/mL	Fenoles mg/mL
A1	77.6 ± 2.34	51.82 ± 5.56	0.86 ± 0.03	0.96 ± 0.04
A2	358.6 ± 4.40	2413.03 ± 3.17	0.82 ± 0.06	1.19 ± 0.07
E1	303.2 ± 3.26	96.4 ± 4.44	1.14 ± 0.22	0.95 ± 0.02
E2	12,606.8 ± 0.51	1073.5 ± 1.07	14.74 ± 1.6	8.56 ± 0.02
E3	13,366.5 ± 7.66	4363.9 ± 4.14	9.21 ± 0.84	8.32 ± 0.18
U1	264.9 ± 4.07	123.6 ± 5.36	1.64 ± 0.17	0.8 ± 0.05
U2	13,525.1 ± 1.72	4933.33 ± 0.71	9.41 ± 0.15	5.33 ± 0.09
U3	641.0 ± 1.40	2127.42 ± 1.07	4.14 ± 0.12	2.19 ± 0.25

Treatments: A1 = hydrodistillation; A2 = hydro + ultrasound; E1 = hexane; E2 = ethanol; E3 = acetone; U1 = hexane + ultrasound; U2 = ethanol + ultrasound; U3 = acetone + ultrasound.

## Data Availability

The original contributions presented in this study are included in the article. Further inquiries can be directed to the corresponding author.

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
