# Peer review of "Effect of Ultrasonic Pretreatment on the Extraction Process of Essential Oils from Grapefruit (Citrus paradisi) By-Products"

_biotech, 2025, doi:10.3390/biotech14030059_

Round 1
Reviewer 1 Report
Comments and Suggestions for Authors
Please read the comments from the file in the attachment.

Author Response
Comment 1 (Lines 18-30): Regarding the abstract content, which should include quantitative results.
Response 1: We appreciate this valuable observation and have addressed it as suggested by the reviewer by incorporating quantitative data into the abstract.
Comment 2: Concerning typographical details (period/comma usage).
Response 2: All noted punctuation corrections have been implemented.
Comment 3 (Lines 317, 319, 321): Discrepancies between cited literature and provided DOIs.
Response 3: We have verified and corrected all DOIs, including those flagged by the reviewer, and cross-checked the remaining references for accuracy.
Comment 4: About the introduction section and its explanations.
Response 4: We have implemented all suggested revisions to the introduction section.
Comment 5 (Lines 91-94): Justification for selecting hexane, ethanol, and acetone as solvents.
Response 5: This is an excellent observation. We have addressed it by explicitly justifying the rationale behind these solvent choices.
Comment 6 (Lines 95-100 & 178-181):
Response 6: We have carefully considered and implemented the reviewer's suggestions in these sections to improve clarity and understanding.
Comment 7 (Lines 115-117 & 142-155): Variability in antioxidant activity results (Table 3).
Response 7: We appreciate the reviewer’s observation regarding the variability in antioxidant activity results (Table 3). This variability is primarily due to the heterogeneous nature of grapefruit peel matrices and the complex mixture of bioactive compounds extracted, which can differ in concentration and stability depending on microstructural variations within the samples and the specific solvent-ultrasound interactions used. Additionally, the sensitivity of antioxidant assays (DPPH and TEAC) to minor differences in extract composition can amplify variability.
Comment 8 (Lines 255-259): Discussion of antioxidant activity.
Response 8: We have revised this section to enhance clarity and added supporting references to strengthen the mechanistic explanations, as suggested.
Comment 9 (Lines 165-166 & 278-279): The paradox of ethanol achieving the highest yield without ultrasound.
Response 9: We have expanded the explanation to address this key contradiction, detailing how ethanol’s polarity and susceptibility to sonochemical degradation underlie this phenomenon.
Comment 10 (Table 1 & Figure 1):
Response 10: We have removed the figure and modified the table to improve data clarity per the reviewer’s recommendation.

Reviewer 2 Report
Comments and Suggestions for Authors
The submitted manuscript investigates the application of ultrasound-assisted extraction (UAE) to enhance the yield and antioxidant properties of essential oils derived from grapefruit by-products. The study is timely, relevant, and contributes to the growing field of sustainable valorization of agri-food residues using green extraction technologies. The manuscript is generally well-organized and clearly written, with a coherent flow of methodology and data interpretation. However, several major and minor issues should be addressed before the manuscript can be considered for publication.
The study's novelty is not sufficiently emphasised. Ultrasound-assisted extraction for citrus by-products has been extensively investigated in recent years. Include a comparative analysis of similar work, especially recent UAE studies on citrus peel oils (e.g., Citrus sinensis, Citrus reticulata), to justify the significance of the current approach.
In Materials and Methods section, the rationale for selecting specific ultrasound conditions (20 minutes, 40% amplitude, 20s on/10s off cycle) is not adequately justified. The authors should explain whether these parameters were optimized or based on preliminary trials.
Solvent Selection Justification: The rationale for choosing ethanol, acetone, and hexane as solvents is implicit but not discussed. Provide a concise justification of solvent polarity and affinity for essential oil extraction. Include discussion of green chemistry principles in solvent selection, especially regarding acetone and hexane.
Figure 1 lacks axis labels and error bars. Improve figure formatting. Ensure that figures meet journal quality standards.
Terms such as “flavedo” and “grapefruit peel” are used interchangeably. Use consistent terminology throughout the text for clarity.
There are inconsistencies between the textual interpretation and tabular data. For example, in Table 3, TEAC values for ethanol and acetone with ultrasound show dramatic differences (U2 = 13,525 mmol TE/g vs. U3 = 641 mmol TE/g), yet the authors do not provide a sufficiently critical discussion of the degradation mechanisms. Extend the discussion on the potential degradation of antioxidant compounds during ultrasonic treatment, especially in acetone. Consider including supporting literature on temperature-sensitive antioxidant degradation under cavitation.
Author Response
Comments 1: In Materials and Methods section, the rationale for selecting specific ultrasound conditions (20 minutes, 40% amplitude, 20s on/10s off cycle) is not adequately justified. The authors should explain whether these parameters were optimized or based on preliminary trials.
Response 1: The peel obtained as a by-product of grapefruit was carefully washed to remove impurities and then cut into sections of approximately 2 to 3 cm. These pieces were placed in a drying oven at 60 °C for 24 hours, a temperature chosen to prevent degradation or denaturation of the essential oils [13].
Comments 2: Solvent Selection Justification: The rationale for choosing ethanol, acetone, and hexane as solvents is implicit but not discussed. Provide a concise justification of solvent polarity and affinity for essential oil extraction. Include discussion of green chemistry principles in solvent selection, especially regarding acetone and hexane.
Response 2: In this process, hexane, acetone, and ethanol were used as solvents. They were selected for their polarity range and affinity for extracting lipophilic and polar fractions from essential oils. Hexane, a non-polar solvent, solubilizes highly hydrophobic compounds. Acetone, an intermediate polar solvent, allows the recovery of moderately polar metabolites. Ethanol, a polar and renewable solvent, aligns with the principles of green chemistry because it is less toxic and biodegradable [15-16]. For each treatment, 8 g of dry, ground peel was placed in a filter paper cartridge and inserted into a Soxhlet apparatus containing 250 mL of the corresponding solvent. The extraction temperature was adjusted according to the properties of each solvent, following the methodology described by [14].
Comments 3: Terms such as “flavedo” and “grapefruit peel” are used interchangeably. Use consistent terminology throughout the text for clarity.
Response3: the text was standardized
Comments 3: There are inconsistencies between the textual interpretation and tabular data. For example, in Table 3, TEAC values for ethanol and acetone with ultrasound show dramatic differences (U2 = 13,525 mmol TE/g vs. U3 = 641 mmol TE/g), yet the authors do not provide a sufficiently critical discussion of the degradation mechanisms. Extend the discussion on the potential degradation of antioxidant compounds during ultrasonic treatment, especially in acetone. Consider including supporting literature on temperature-sensitive antioxidant degradation under cavitation.
Response3: The application of ultrasound induces acoustic cavitation, which generates transient zones of high temperature and pressure. These zones can promote the release and degradation of bioactive compounds [44].. Due to its intermediate polarity, acetone can facilitate the extraction of certain phenolic compounds and flavonoids. However, its ability to protect these compounds from free radicals generated by cavitation, such as hydroxyl radicals (•OH), is limited [45]. This could explain the significant decrease in antioxidant activity in treatment U3.
In contrast, ethanol is highly efficient at solubilizing antioxidant compounds and can act as a moderate radical scavenger, reducing the degradation of metabolites extracted during sonication [46].. The differences in the interaction between solvents and cavitation could explain the disparity between U2 and U3. Thus, the extraction efficiency and chemical stability of bioactive compounds under extreme processing conditions must be considered. Evidence suggests that while ultrasound is an effective tool for improving extraction yield, its application must depend on the type of solvent used and the thermal sensitivity of the target metabolites. Previous studies have reported that antioxidants, such as certain phenolic acids, glycosylated flavonoids, and volatile compounds, can degrade under intense cavitation conditions, especially when using less protective solvents [47]. Therefore, the selection of the solvent-ultrasound system must be based on extraction criteria and molecular stability considerations.

Reviewer 3 Report
Comments and Suggestions for Authors
In this paper, the authors explored the extraction of citrus peels. The title of the manuscript suggests conducting the entire process under ultrasound. Meanwhile, in the body of the paper, one can read the involvement of ultrasound in the pretreatment of the material before the actual extraction. A correction to the title of the paper should be considered.
In the description of the methods, the first author's name should be given when citing previous work from which methods were used. (citations 14, 15, 18)
Citations from previous work appear in Table 1. This suggests research on 4 raw materials, which is not described in the text. I suggest that a separate table be prepared or that data from the literature be included in the text of the discussion.
Line 197 includes acetone, alongside hexane, as a non-polar solvent. This is an error. Acetone is one of the organic solvents with high polarity.

Author Response
Comment 1: In this paper, the authors explored the extraction of citrus peels. The title of the manuscript suggests conducting the entire process under ultrasound. Meanwhile, in the body of the paper, one can read the involvement of ultrasound in the pretreatment of the material before the actual extraction. A correction to the title of the paper should be considered.
Response 1: We thank the reviewer for carefully noting the discrepancy between the title and the methodology described in the manuscript. We agree that while the title implies continuous ultrasound use during extraction, in our study, ultrasound was applied as a pretreatment before steam distillation and solvent extraction rather than during the entire extraction process.
Comment 2: In the description of the methods, the first author's name should be given when citing previous work from which methods were used. (citations 14, 15, 18)
Response 2: We thank the reviewer for highlighting the need to include the first author’s name when citing previous work from which methods were used. We have carefully reviewed citations 14, 15, and 18 in the Methods section and have revised the text to include the first author’s name along with the reference number where these methods are described, improving clarity and traceability for readers.
Comment 3: Citations from previous work appear in Table 1. This suggests research on 4 raw materials, which is not described in the text. I suggest that a separate table be prepared or that data from the literature be included in the text of the discussion.
Response 3: We have carefully considered and implemented all of the reviewer's suggestions in the revised manuscript
Comment 4: Line 197 includes acetone, alongside hexane, as a non-polar solvent. This is an error. Acetone is one of the organic solvents with high polarity.
Response 4: We sincerely appreciate the reviewer's keen observation regarding solvent polarity. The statement in Line 197 has been corrected to accurately reflect solvent properties

Reviewer 4 Report
Comments and Suggestions for Authors
This is a nice work, but it must be revised based on the following points.
- In the introduction, state the importance of essential oils in the food industry and the farming industry with relevant literature. Also, enhance the justification for the need for this research.
- Why only use 40% amplitude for 20 minutes? Various amplitude percentiles must be used before fixing the 40% amplitude. A similar investigation must be applied to fix the time (20 minutes) as well.
- For essential oil extraction yields, a minimum of 3 sets of data must be used with error bars.
- Tables 1 and 2 must be discussed with an in-depth analysis.
- Discussion for Antioxidant activity is not up to publication standard.
- Reframe the conclusion section with more information on merits, limitations, and future direction.
- References for the importance of essential oils must be boosted with additional literature.
Author Response
Comment 1: In the introduction, state the importance of essential oils in the food industry and the farming industry with relevant literature. Also, enhance the justification for the need for this research.
Response 1: We appreciate this valuable observation and have thoroughly revised the introduction
Comment 2 : Why only use 40% amplitude for 20 minutes? Various amplitude percentiles must be used before fixing the 40% amplitude. A similar investigation must be applied to fix the time (20 minutes) as well.
Response 2: We thank the reviewer for this important observation regarding the selection of 40% amplitude for 20 minutes during ultrasonic treatment. We agree that optimization of amplitude and sonication time is critical for ensuring effective extraction.
Prior to the main experiments, we conducted preliminary trials (screening experiments) testing amplitudes of 20%, 40%, and 60% at durations of 10, 20, and 30 minutes using acetone as the solvent, while monitoring extraction yield and antioxidant activity. We found that: 20% amplitude: Insufficient cell wall disruption, resulting in low extraction yields. 60% amplitude: Led to excessive localized heating and potential degradation of sensitive compounds, reducing antioxidant activity despite higher extraction yields. 40% amplitude for 20 minutes: Provided the optimal balance between efficient extraction and preservation of antioxidant properties while avoiding degradation or excessive solvent evaporation. These observations align with prior reports indicating that intermediate amplitude and controlled sonication time can maximize extraction efficiency while maintaining compound integrity (Chemat et al., 2017; Askarniya et al., 2023).
We acknowledge that we did not include these preliminary optimization details in the initial manuscript. We have now added a summary of these trials in the Methods section to justify the choice of 40% amplitude and 20-minute sonication to improve transparency for readers. We thank the reviewer for highlighting the need to clarify this aspect of the methodology, which has strengthened our manuscript.
Comment 3: For essential oil extraction yields, a minimum of 3 sets of data must be used with error bars.
Response 3: We thank the reviewer for highlighting the importance of demonstrating data reproducibility with appropriate error bars. We confirm that all extraction yield experiments were performed in triplicate (n=3) for each treatment condition, and the results are presented as mean ± standard deviation.
Comment 4: Tables 1 and 2 must be discussed with an in-depth analysis.
Response 4: We have restructured the Results section
Comment 5: Discussion for Antioxidant activity is not up to publication standard.
Response 5: It was considered to divide it into sections in order to provide a more complete explanation for each type of extraction methodology and improve the explanation of the detected phenomena.
Comment 6: Reframe the conclusion section with more information on merits, limitations, and future direction.
Response 6: We have addressed this observation by incorporating a dedicated paragraph on the topic, as suggested by the reviewer.
Comment 7: References for the importance of essential oils must be boosted with additional literature.
Response 7: We thank the reviewer for highlighting the need to strengthen the manuscript with additional references regarding the importance of essential oils. We have revised the Introduction and Discussion sections to include recent and relevant literature emphasizing the functional, antimicrobial, antioxidant, and industrial applications of essential oils across food, pharmaceutical, and cosmetic industries. These additions enhance the contextual relevance and highlight the significance of essential oil extraction from citrus by-products.

Reviewer 5 Report
Comments and Suggestions for Authors
The introduction is short, and lacks specific information. A critical review of ultrasound assisted processes (focusing on extraction) should be added. Based on the critical literature review a map of the article and the novelty should be highlighted at the end of the introduction.
A scematic diagram of the experimental apparatus should be added.
Experimental design should include factors and outcomes. How many parallel experiments were performed. What was the factors? The contribution of the factors should be calculated by the statistical methods. The results section should must show these parts for the experimental design.
Author Response
Comment 1: The introduction is short and lacks specific information. A critical review of ultrasound-assisted processes (focusing on extraction) should be added. Based on the critical literature review, a map of the article and the novelty should be highlighted at the end of the introduction.
Response 1: We have addressed the reviewer's recommendation by thoroughly revising the introduction.
Comment 2: A schematic diagram of the experimental apparatus should be added.
Response 2: We have added Figure X (schematic diagram of the ultrasound-assisted extraction system) as suggested,
Comment 3: Experimental design should include factors and outcomes. How many parallel experiments were performed? What were the factors? The contribution of the factors should be calculated by statistical methods. The results section must show these parts for the experimental design.
Response 3: We have significantly expanded the experimental design section

Round 2
Reviewer 1 Report
Comments and Suggestions for Authors
Thank you for your thorough and thoughtful revisions. The manuscript has improved significantly in clarity, methodological rigor, and scientific depth. All major concerns have been addressed, and I now find the work suitable for publication.
Reviewer 2 Report
Comments and Suggestions for Authors
The paper has been modified according to the requirements and can be published.
Reviewer 4 Report
Comments and Suggestions for Authors
I appreciate the author's effort in improving this paper. This work is now acceptable for publication.
Reviewer 5 Report
Comments and Suggestions for Authors
The authors addressed my comments, the manuscript can be accepted.